# Caenorhabditis elegans models for striated muscle disorders caused by missense variants of human LMNA

Ellen F. Gregory[1], Shilpi Kalra[1], Trisha Brock[2], Gisèle Bonne[3], G. W. Gant Luxton[1], Christopher Hopkins[2], Daniel A. Starr[1]*

**1** Department of Molecular and Cellular Biology, University of California, Davis, California, United States of America, **2** InVivo Biosystems, Eugene, Oregon, United States of America, **3** Sorbonne Université, Inserm, Institut de Myologie, Centre de Recherche en Myologie, Paris, France

* dastarr@ucdavis.edu

**Data Availability Statement:** All the data are reported in the manuscript.

## Abstract

Striated muscle laminopathies caused by missense mutations in the nuclear lamin gene *LMNA* are characterized by cardiac dysfunction and often skeletal muscle defects. Attempts to predict which *LMNA* variants are pathogenic and to understand their physiological effects lag behind variant discovery. We created *Caenorhabditis elegans* models for striated muscle laminopathies by introducing pathogenic human *LMNA* variants and variants of unknown significance at conserved residues within the *lmn-1* gene. Severe missense variants reduced fertility and/or motility in *C. elegans*. Nuclear morphology defects were evident in the hypodermal nuclei of many lamin variant strains, indicating a loss of nuclear envelope integrity. Phenotypic severity varied within the two classes of missense mutations involved in striated muscle disease, but overall, variants associated with both skeletal and cardiac muscle defects in humans lead to more severe phenotypes in our model than variants predicted to disrupt cardiac function alone. We also identified a separation of function allele, *lmn-1(R204W)*, that exhibited normal viability and swimming behavior but had a severe nuclear migration defect. Thus, we established *C. elegans* avatars for striated muscle laminopathies and identified *LMNA* variants that offer insight into lamin mechanisms during normal development.

## Author summary

Muscular dystrophy is a progressive muscle-wasting disorder that eventually leads to cardiac disease. Mutations in the *LMNA* gene, which encodes an intermediate filament protein involved in the structure and organization of the nucleus, is a common but poorly understood cause of this disease. How variants across the breadth of *LMNA* contribute to mechanistic cellular defects that lead to disease is poorly understood, leading to hurdles in diagnosing disease and developing treatments. We found that by introducing amino acid substitutions found in patients with striated muscle disorders caused by *LMNA* into the conserved *lmn-1* gene of the nematode *C. elegans*, we could rapidly test the function of

**Funding:** These studies were supported by the National Institutes of Health grants R35GM134859 to D.A.S. and R01GM129374 to G.W.G.L., and Agence Nationale de la Recherche grant #ANR-19-CE17-0013-01 to G.B. The funders had no role in study design, data collection and analysis, decision to publish, or preparation of the manuscript.

**Competing interests:** The authors have declared that no competing interests exist.

these variants to better understand their roles. We found that variants modeling diseases that involve both skeletal and cardiac muscle in humans were the most pathogenic in *C. elegans*, typically affecting both viability and movement, while those that modeled cardiac disease alone had less deleterious effects in *C. elegans*. Thus, our new *C. elegans* models can be used to diagnose and predict the severity of new variants of human *LMNA* as well as to better understand the molecular mechanisms of lamins in normal development.

## Introduction

Lamins are highly conserved intermediate filament proteins that underlie the inner nuclear membrane of the nuclear envelope and form a nucleoskeletal meshwork known as the nuclear lamina [1]. Lamins are integral to the mechanical stability of the nucleus, cytoskeletal coupling, genome organization, and gene expression [2]. Pathogenic variants of the human *LMNA* gene contribute to a broad spectrum of tissue-specific diseases that are collectively referred to as laminopathies [3,4]. 80% of pathogenic mutations in *LMNA* give rise to striated muscle laminopathies [5].

Striated muscle laminopathies lead to dilated cardiomyopathy (DCM-CD), which is the main cause of death in affected individuals [3]. However, the range and severity of symptoms within any given *LMNA*-associated striated muscle disease varies [5–7]. While many individuals only have cardiac defects [8], others additionally experience impaired skeletal muscle function [9,10]. At the severe end of the spectrum, *LMNA*-related congenital muscular dystrophy (L-CMD) presents as early onset dystrophic symptoms and rapid disease progression [9,11]. Autosomal-dominant Emery-Dreifuss muscular dystrophy (AD-EDMD) is characterized by childhood onset and manifests as gradual progressive skeletal muscle weakness [6,12]. In contrast, limb-girdle muscular dystrophy type 1B (LGMD1B) is a primarily adult-onset disease with milder weakness typically affecting proximal muscle [13]. How mutations within *LMNA* lead to multiple diseases with a wide range of overlapping etiologies is poorly understood.

Striated muscle laminopathies lack a clear phenotype-genotype link, which further complicates prognoses. The location of a disease-associated variant within the *LMNA* gene does not correspond to symptom onset, type, or severity [14,15]. The need for precise characterization of known pathogenic *LMNA* mutations is compounded by the accelerated pace of variant discovery, which has identified hundreds of variants of unknown clinical significance (VUS), most commonly missense variants [16]. Therefore, a model that can efficiently evaluate the pathogenicity of specific variants is critical to accelerate the diagnosis of *LMNA* VUS. Furthermore, new models would expand our understanding of both the physiological progression of laminopathies and basic roles of lamins throughout normal development.

Vertebrate models have been developed to study the effects of several laminopathic variants. Zebrafish containing a 5-bp deletion that is predicted to create an early stop codon in one or both copies of *lmna* have skeletal muscle defects, reduced movement, and aberrant expression of genes that are also dysregulated in laminopathy patients [17]. $Lmna^{-/-}$ mice lose nuclear envelope integrity, have delayed postnatal growth, and experience a rapid onset of muscular dystrophy, while $Lmna^{+/-}$ mice develop adult-onset DCM [18]. Murine models also exist for laminopathic missense mutations. A line designed to model AD-EDMD with the $Lmna^{L530P/L530P}$ mutation produces mice with symptoms similar to premature aging disorders [19]. Another mouse model, $Lmna^{H222P/H222P}$, develops the dystrophic and cardiac phenotypes of AD-EDMD [20,21]. Notwithstanding these advances, vertebrate models are time-consuming

and expensive to generate, limiting their ability to encompass the spectrum of *LMNA*-associated striated muscle laminopathies.

The model nematode *Caenorhabditis elegans* is genetically and microscopically tractable, making it suitable for quickly generating and characterizing mutant strains and an excellent system for studying striated muscle laminopathies. *C. elegans*, like other invertebrates, has a single lamin protein, LMN-1, which is both a strength and weakness of our approach. LMN-1 was previously termed a B-type lamin because it is constitutively farnesylated and expressed in undifferentiated tissues [22]. Although, LMN-1 processing is less complicated than human laminA, LMN-1 is equally related to vertebrate lamins A, B1, and B2 [23]. Furthermore, LMN-1 performs many of the same functions as the vertebrate lamins. LMN-1 binds BAF, emerin, and other LEM-domain homologs, and knock-down of *lmn-1* leads to defects in nuclear shape, chromosome segregation, nuclear import, germline organization, and embryonic viability [22,24–30]. Because many of the residues in human lamin A that are implicated in disease are conserved in LMN-1, *C. elegans* has been used to model laminopathy-associated missense variants, and these animals exhibit phenotypes such as defects in striated body-wall muscles that are reminiscent of human pathologies [31–35].

Despite these important advantages, *C. elegans* models to date have been limited in their ability to recapitulate human laminopathies. Many published models express missense variant LMN-1 proteins from multi-copy extrachromosomal arrays in the presence of the intact endogenous *lmn-1* locus [31–35]. Consequently, the relative levels of mutant and wild-type LMN-1 are difficult to accurately measure and may vary from tissue to tissue or cell to cell. In addition, most models fuse LMN-1 to fluorescent proteins or tags that have been demonstrated to disrupt lamin function *in vivo* [31,34,36]. One model utilized an untagged, single-copy of LMN-1(N209K), which is likely to better model the human disease. However, analyses of this strain focused on the early embryo [37].

Our goal is to characterize laminopathy-associated missense variants of human *LMNA* in order to accelerate phenotypic evaluation of known pathogenic variants and emergent VUS. We established new models that express missense variant proteins from the endogenous *C. elegans lmn-1* locus to better model human disease. We selected *lmn-1* mutations that are homologous to *LMNA* missense variants reported in striated muscle laminopathy patients and classified them based on whether they were known to exhibit cardiac defects only (DCM-CD), or both cardiac and skeletal muscle defects (AD-EDMD). We also evaluated four *LMNA* VUS to determine their potential pathogenicity. To functionally characterize lamin variants, we assayed animal viability and motility. Lamins also interact with components of the LINC complex, which is required to move nuclei during cell migration and has been implicated in striated muscle disorders. We therefore quantified nuclear migration and morphology in hypodermal tissue, revealing a separation of function allele that could provide insights into lamin function during nuclear migration.

## Results

### Introduction of human *LMNA* variants linked to skeletal and cardiac muscle laminopathies into *C. elegans lmn-1*

Our goal was to take clinical *LMNA* variants and edit them into the *C. elegans lmn-1* gene. We selected eight missense variants throughout the open reading frame of human *LMNA* that are linked to diseases that affect skeletal and/or cardiac muscle (Fig 1A). The diseases associated with each selected missense variant are listed in Fig 1A. Three mutations in human *LMNA* (p. E82K, p.E161K, and p.R190W, corresponding to E96K, E175K, and R204W in *C. elegans* LMN-1, respectively) cause defects primarily in cardiac muscle [38–53]. Five variants in

**Fig 1. Introducing Human *LMNA* Missense Variants into *C. elegans lmn-1*.** (A) Table of human *LMNA* missense variants and the homologous *C. elegans* LMN-1 residue changes. Variants are colored according to their clinical classification: missense variants that exhibit both cardiac and skeletal muscle defects in humans (red), variants that affect cardiac muscle specifically (yellow), and VUS (blue). Clinical presentations and the number of subjects reported are from www.umd.be/LMNA/ (and G Bonne, R Ben Yaou personal communication). L-CMD: *LMNA*-related congenital muscular dystrophy, EDMD: Emery-Dreifuss muscular dystrophy, LGMD1B: limb-girdle muscular dystrophy type 1B, DCM-CD: dilated cardiomyopathy with conduction defects, SML: striated muscle laminopathy (subtype undetermined). Asymptomatic subjects are essentially young 'DCM family' members who may have not yet developed cardiac disease. (B) Alignment of *C. elegans* LMN-1 and human lamin A/C, lamin B1 and lamin B2 proteins generated by Clustal-Omega multiple sequence alignment. The amino acid changes examined in this study are indicated with boxes, colored as in 1A. *C. elegans* residues are listed above the boxes, and the corresponding human residues are shown below. (C) Diagram of the *C. elegans* LMN-1 protein structure with the head (orange), coils (red) and tail (blue) domains indicated. The positions of the amino acid changes featured in this study are shown using the same color scheme as in A.

human *LMNA* (p.N39S, p.Y45C, p.R50P, p.E358K and p.L530P, equivalent to N53S, Y59C, R64P, E358K, and L535P in LMN-1), are associated with earlier age of symptom onset and often are considered more severe, as they contribute to diseases that affect both cardiac and skeletal muscles (red throughout the figures) [6,9,10,12,14,54–60].

We also chose missense *LMNA* VUS (p.K270Q, p.R331Q, and p.G523R, corresponding to K284Q, K331Q, and G528R in LMN-1, respectively) that are suspected to be involved in disease. *LMNA* p.G523R has a relatively high allele frequency (0.00006), according to the Genome Aggregation Database (gnomAD) [61], and is therefore predicted to be benign. However, the ClinVar database [62] designates p.G523R, as well as two other *LMNA* VUS (p.K270Q and p. R331Q), as potentially pathogenic based on clinical testing, large-scale genomic analyses of patient cohorts, and *in silico* structural predictions (Fig 1A) [14,51,63–69]. *LMNA* pR331Q is associated with DCM-CD [68], but remains annotated in the ClinVar database [62] as a variant with conflicting interpretations of pathogenicity. We therefore categorized pR331Q as a VUS for the purposes of this study. The fourth VUS, *LMNA* p.S407D, which corresponds to LMN-1 (G407D), is a predicted benign variant that has not previously been associated with disease [70].

We generated homologous missense variants of *C. elegans lmn-1* corresponding to each of the eight pathogenic *LMNA* variants and the four VUS (Fig 1). Homozygous point mutations were engineered into the endogenous *lmn-1* locus using CRISPR-Cas9-mediated genome editing. We will refer to the variants using the amino acid position in the *C. elegans* LMN-1 protein throughout the rest of this manuscript.

### *lmn-1* variants that model skeletal and cardiac muscle disorders exhibit reduced fitness

*lmn-1 (RNAi)* causes embryonic lethality [22]. Chromosome segregation defects are evident as early as the 2-cell stage, and embryos arrest around the 100-cell stage, indicating that *lmn-1* is required for early embryonic development [22]. Although *lmn-1* null homozygous animals from heterozygous mothers are able to develop with the help of maternally-loaded LMN-1, they are sterile due to the lack of lamin in the germline [71]. We therefore examined the effect of the *lmn-1* missense mutations on overall *C. elegans* health and fitness by quantifying the brood size and level of embryonic lethality of each engineered line.

All five *C. elegans* strains carrying missense *lmn-1* mutations that model human *LMNA* variants associated with both skeletal and cardiac muscle dysfunction had significant viability and fertility defects (Fig 2). Homozygous N53S, Y59C, R64P, and L535P LMN-1 variants had the most deleterious effects on brood size. These strains were maintained as balanced heterozygotes. Homozygous animals from heterozygous mothers survived to adulthood, which allowed us to assay the effect of having no genetically-encoded wild-type copies of *lmn-1*. Homozygous L535P adults were sterile, failing to lay any eggs. N53S homozygous animals laid some eggs, but none were viable. LMN-1 Y59C and R64P homozygous animals had small broods, roughly 30–40% the size of those of wild type animals. In addition, embryonic lethality in Y59C and R64P animals was 20–30% higher than that of wild type. E358K also had a significantly reduced brood size, less than 40% that of wild type. Half of E358K animals laid few or no embryos and had elevated lethality. In contrast to the mutants modeling diseases affecting both skeletal and cardiac muscles, lmn-1 mutants modeling LMNA variants associated solely with DCM-CD (E96K, E175K, and R204W), did not show pronounced reductions in brood size or heightened lethality. Likewise, the four VUS showed no defects in viability or fertility compared to wild type animals. These results indicate that our *C. elegans* models reflect how missense variants linked to muscle and cardiac laminopathies like L-CMD and EDMD in humans reduce overall health and fitness.

### Some *lmn-1* missense mutants had impaired swimming behavior

Muscular dystrophies usually manifest as defects in skeletal muscle function that variously restrict mobility [5]. We therefore assayed swimming behavior [32,72] in our *lmn-1* variant lines to test whether physiological changes in *C. elegans* might predict the potential severity of

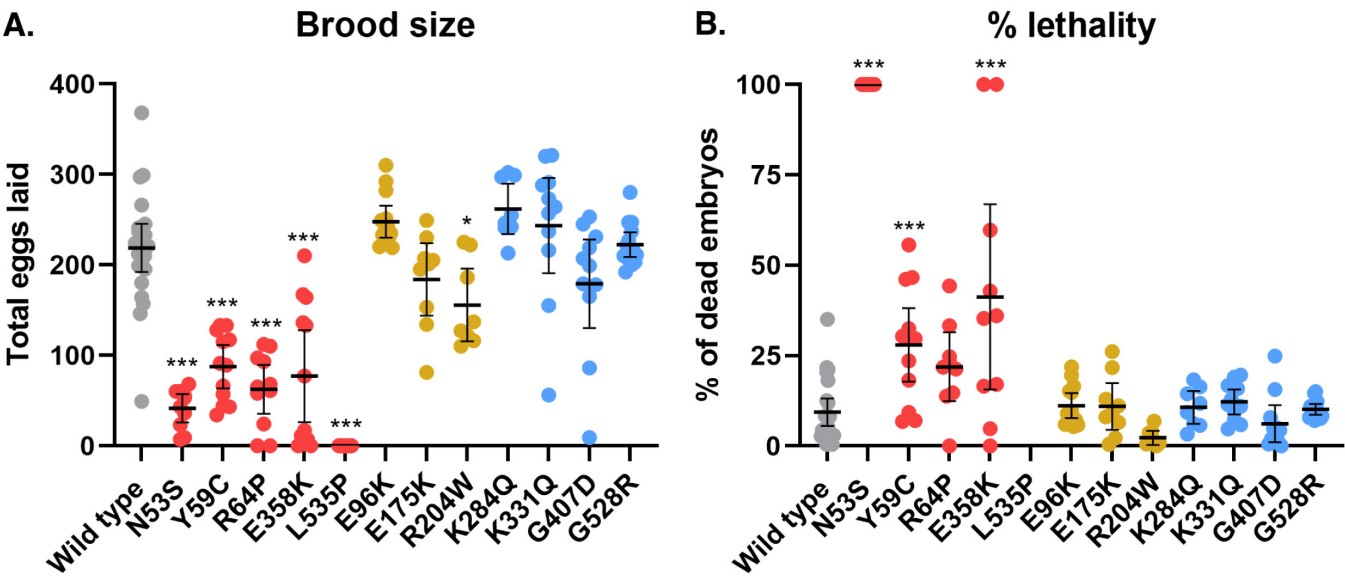

**Fig 2. Severe Human Pathogenic *LMNA* Variants Modeled in *C. elegans lmn-1* Cause Decreased Brood Size and Elevated Embryonic Lethality.** (A) Brood size of wild type and *lmn-1* homozygous missense mutant lines. Each dot represents the average number of eggs laid by a single animal. (B) The percent of embryos laid that failed to hatch within 24 hours for each genotype is shown. Each dot represents the average number of eggs laid by a single animal. Both A and B are grouped and colored according to clinical classifications as in Fig 1. n = 22 wild type animals, and n = 8–12 in all other strains. Means and 95% CI are shown. Significance compared to wild type was calculated using a one-way ANOVA and corrected for multiple comparisons using Dunnett's test. *p≤0.05; ***p≤0.001.

*LMNA*-associated striated muscle disease missense variants. We assayed major motor movements by observing animals thrashing in buffer and quantified the number of body bends per second (BBPS) for each of the homozygous missense lines (Fig 3). We also measured swimming in heterozygous animals carrying variants that caused major viability and fertility defects.

LMN-1 L535P homozygous animals had the most severe motility defects, averaging less than 1 BBPS, followed by LMN-1 Y59C with a rate of 1.1 BBPS compared to 1.86 BBPS in wild type (Fig 3B). However, we observed a wide distribution of phenotypes within each strain, with individual animals exhibiting anywhere from normal swimming behavior to severely reduced or undetectable movement. To test the significance of the effect of *lmn-1* missense mutations on motility, we scored the percentage of individuals in each strain that thrashed more slowly than the mean rate of *lmn-1(Y59C)*, which at 1.1 BBPS is a well-studied mutation in *C. elegans* known to disrupt swimming (Fig 3C) [31,32,72].

Four homozygous *lmn-1* mutations (Y59C, R64P, K331Q and L535P) caused severe swimming defects. Three of these mutations (Y59C, R64P, and L535P) were designed to model human mutations in *LMNA* that affect both skeletal and cardiac muscle function. Of the alleles that were designed to model cardiomyopathy-linked variants without skeletal muscle involvement (Fig 3), none caused significant motility defects. Three of the VUS, K284Q, G407D, and G528R, had normal swimming rates (Fig 3B and 3C). In contrast, K331Q had a statistically significant swimming defect, suggesting this VUS is pathogenic. K331 has been linked to DCM-CD (Fig 1A and [68]), validating our approach for predicting VUS pathogenicity.

## A cardiomyopathy-linked missense variant drastically disrupts nuclear migration

Knocking down *lmn-1* via RNAi leads to embryonic lethality [22]. However, animals that survive, potentially due to higher residual protein levels, have nuclear migration defects in

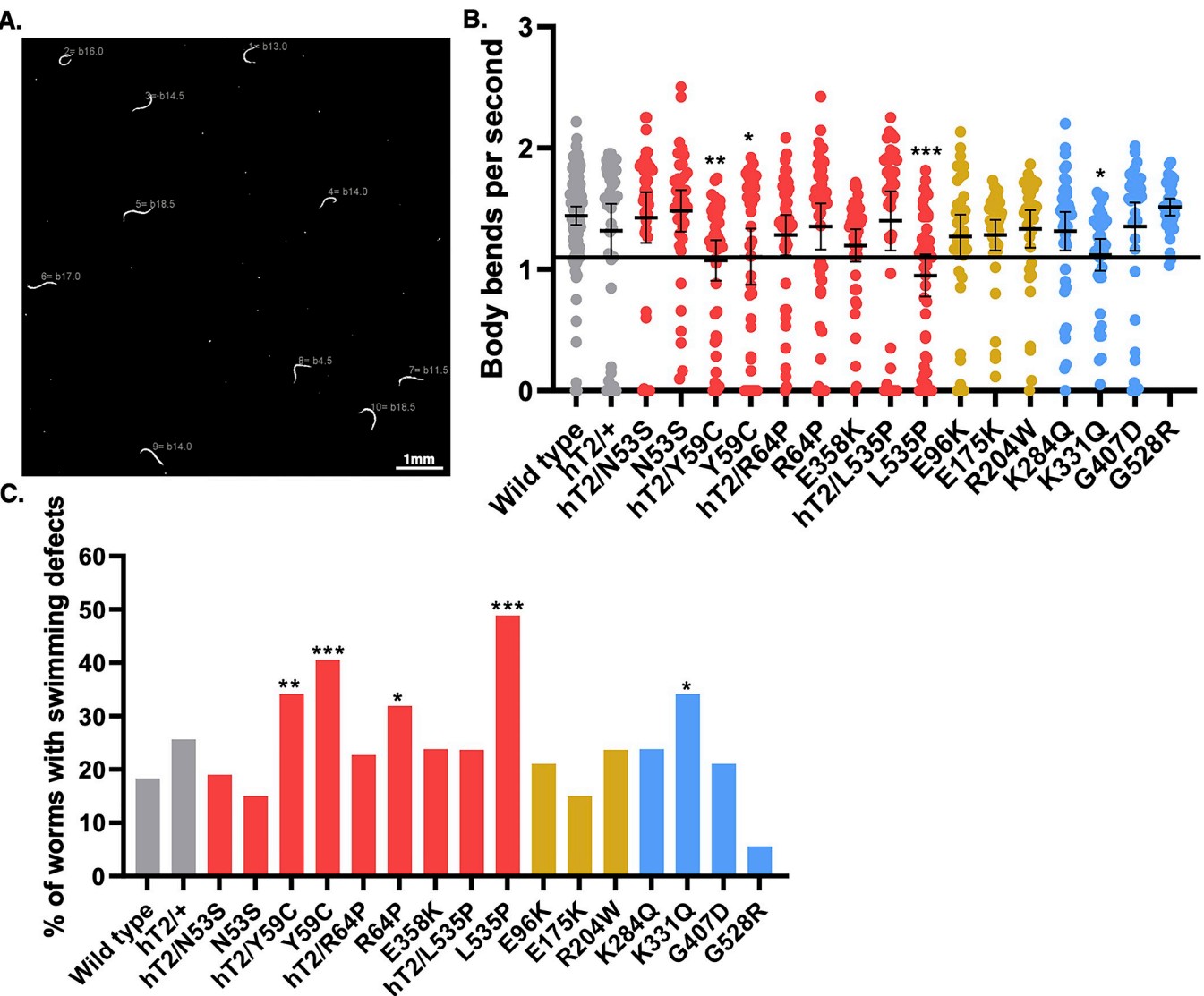

**Fig 3. *lmn-1* Missense Variants can Reduce Swimming Motility.** (A) An example of a processed swimming video. Each animal is assigned a number by the Fiji wrMTrck plugin, which records the number of body bends per second (BBPS) (shown in dark gray beside the animals). (B) Each dot represents a single animal that was tracked over 30 seconds. BBPS over the 30 second interval is shown. Genotypes (homozygous (e.g., N53S) and heterozygous LMN-1 mutants over the *hT2* balancer (e.g., hT2/N53S)) are grouped by variant clinical classification as in Fig 1. Significance was calculated using a one-way ANOVA and corrected for multiple comparisons using Dunnett's test. *p≤0.05; **p≤0.01; ***p≤0.001 A threshold of 1.1 BBPS is indicated by the line. Means and error bars of 95% CI are shown. n ~ 40 animals, except for wild type which is ~100. (C) The same data as in B, but instead, the bars represent the percentage of animals with swimming defects below 1.1 BBPS for each strain. Significance was determined by chi-squared test of the percentage of individuals in each strain that fell at or below 1.1 BBPS as compared to wild type. *p≤0.05; **p≤0.01; ***p≤0.001.

embryonic dorsal hyp7 precursor cells [73]. Hyp7 precursor nuclei that fail to migrate are mislocalized in the dorsal cord of *lmn-1(RNAi)* early larval stage animals, suggesting that LMN-1 is required for this migration event (Fig 4A). Therefore, we examined the extent to which our *lmn-1* missense mutations might affect nuclear migration. We used a previously described line expressing a nuclear GFP marker specifically in the hypodermis to assay nuclear migration in hyp7 precursor cells [74]. *lmn-1(N53S)* and *lmn-1(Y59C)* homozygous animals had significant nuclear migration defects, where 20–35% of hyp7 nuclei were mislocalized in the dorsal cord (Fig 4B and 4C). Interestingly, R204W displayed a severe hyp7 nuclear migration defect,

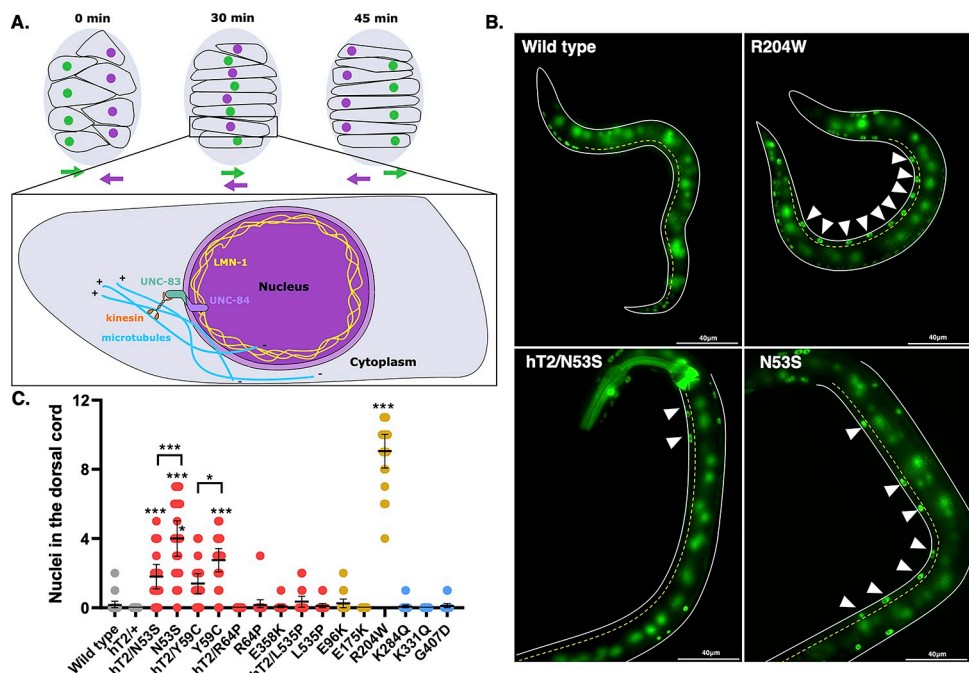

**Fig 4. *lmn-1* Missense Variants can Lead to Defects in Nuclear Migration.** (A) Schematic of nuclear migration in hypodermal cells. Dorsal hyp7 precursor nuclei (purple and green circles) migrate to opposite sides of the embryo (gray; dorsal view; anterior is up) over the course of ~45 minutes. Colored arrows show the direction of migrating nuclei of the same color. Inset shows the molecular motor kinesin (orange) attached to the LINC complex, made up of UNC-83 and UNC-84 (teal and violet, respectively), which binds to lamin (yellow) and facilitates nuclear migration in hyp7 precursors. Microtubules (blue) are shown with their plus ends in front of the migrating nucleus. (B) Representative epifluorescence images of animals with the indicated genotypes expressing a GFP nuclear marker in hyp7 cells. GFP fluorescence in the pharynx of hT2/N53S marks the presence of the hT2 balancer chromosome with a wild-type copy of LMN-1. On the left is the dorsal cord, (yellow dotted line), and the ventral side of the animal is shown on the right. Arrowheads indicate mislocalized nuclei in the dorsal cord. Scale bar: 40μm. (C) Number of nuclei mislocalized in the dorsal cord of wild type and missense variant lines. hT2 designates the presence of a balancer chromosome with a wild-type copy of LMN-1. Each dot represents a single worm. n = 20 per genotype. Means and 95% CI are shown. Significance was calculated using a one-way ANOVA and corrected for multiple comparisons using Tukey's test. *p≤0.05; ***p≤0.001.

where about 75% of hyp7 precursor nuclei failed to migrate (Fig 4B and 4C). None of the other missense mutant lines, including the VUS, significantly disrupted nuclear migration. These data suggest that residues N53 and Y59 contribute to the function of LMN-1 in nuclear migration while R204 is critical to the role of LMN-1 during nuclear migration in the developing hypodermis.

LMN-1 directly interacts with the nucleoplasmic domain of the inner nuclear membrane SUN (Sad1/UNC-84) protein UNC-84, and is necessary for the assembly of the Linker of the Nucleoskeleton and Cytoskeleton (LINC) complex [73]. Once UNC-84 is recruited to the inner nuclear membrane, it recruits the KASH (Klarsicht/ANC-1/SYNE homology) protein UNC-83 to the outer nuclear membrane, and the binding of both proteins in the perinuclear space forms the nucleocytoskeletal bridge known as the LINC complex [75]. To determine whether the *lmn-1(R204W)* variant disrupts the interaction between the LINC complex and LMN-1, we examined the nuclear envelope localization of UNC-83, UNC-84, and LMN-1 (Fig 5). We did not detect a significant change in LMN-1 localization (Fig 5B). In contrast, in the *lmn-1(R204W)* background, the intensity of UNC-83 immunofluorescence was significantly lower than that of wild type in embryonic hyp7 precursor nuclei (Fig 5A and 5B).

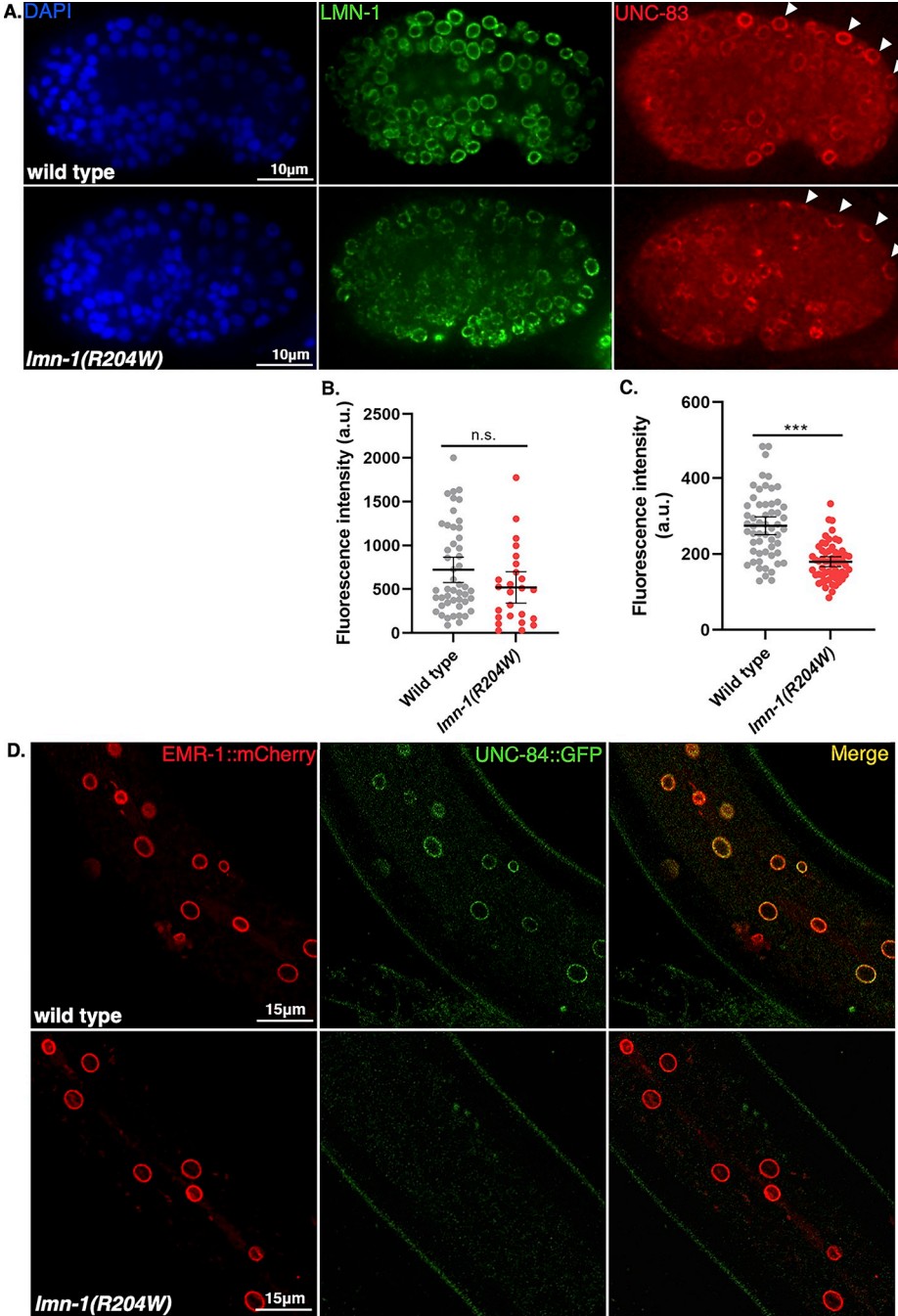

**Fig 5. The *lmn-1(R204W)* Missense Variant Disrupts UNC-84 and UNC-83 Localization to the Nuclear Envelope.**
(A) Immunofluorescence images showing anti-UNC-83 localization to the nuclear periphery of dorsal hypodermal nuclei (white arrowheads) in wild-type and *lmn-1(R204W)* comma-stage embryos. Anti-LMN-1 immunostaining (green) and DAPI-stained nuclei (blue) are also shown. Scale bar: 10μm. (B) Fluorescence intensity (arbitrary units) of anti-LMN-1 immunostaining at the nuclear envelope of wild-type embryos (n = 10) and *lmn-1(R204W)* embryos (n = 12) and (C) Fluorescence intensity (arbitrary units) of anti-UNC-83 immunostaining at the nuclear envelope of wild type embryos (n = 56 nuclei) and *lmn-1(R204W)* embryos (n = 57 nuclei), respectively. Means and error bars of 95% CI are shown. Significance was calculated using student's t test. *** = p<0.001. (D) Representative confocal images of EMR-1::mCherry and UNC-84::GFP-tagged hypodermal nuclei in live wild type and *lmn-1(R204W)* young adult animals. Scale bar: 15μm.

Similarly, we observed reduced fluorescence intensity of GFP-tagged UNC-84 at the nuclear envelope in the hypodermis of R204W compared to wild-type young adult animals (Fig 5C). Together, these data suggest that the LMN-1 R204W variant interferes with the formation of the LINC complex by disrupting UNC-84 and UNC-83 localization to the nuclear envelope in the *C. elegans* hypodermis.

## Some *lmn-1* variants have abnormal nuclear morphology

Disrupting A-type nuclear lamins in mammalian or invertebrate models leads to nuclear blebbing, chromosome bridges, and the formation of micronuclei [22,76–79]. We assayed nuclear morphology in *C. elegans* hypodermal syncytia to determine the extent to which missense mutations in *lmn-1* affect nuclear architecture and stability.

We screened for abnormal nuclear morphology in the hyp7 syncytia of young adults using a soluble nuclear GFP marker expressed in the hypodermis [74]. Three *lmn-1* variant strains (N53S, Y59C, and L535P) had severe nuclear morphology defects, including misshapen and lobulated nuclei and nuclear blebs (Fig 6A). To quantify these defects, we focused on the number of nuclear blebs present in each animal. N53S, Y59C, and L535P animals had the highest number of blebbed nuclei. Strains with the other two variants designed to model both skeletal and cardiac pathologies (R64P and E358K) had fewer, but statistically more than wild type (Fig 6B). R204W, K284Q, and K331Q variant strains also had mild, but statistically significant, nuclear morphology defects. The evidence that missense mutations in *lmn-1* can cause defects in nuclear envelope morphology and nuclear blebs indicates that nuclear instability may play a role in the pathology of striated muscle laminopathies.

## Discussion

Mutations in *LMNA* give rise to a host of diseases that affect striated muscle, which lead to dilated cardiomyopathy that may or may not be associated with skeletal muscle defects. However, symptom onset, type, and severity overlap across striated muscle laminopathies and range widely within each. Furthermore, causative *LMNA* missense variants are not exclusive to specific diseases or domains of the protein, and their effects on lamin structure and function are unclear [15]. The vast majority of known pathogenic *LMNA* mutations are missense variants, and many other variants remain uncharacterized or have conflicting evidence for pathogenicity. Here we establish a pipeline to rapidly generate models of clinical VUS using gene-edited versions of *C. elegans lmn-1*. We found that by subjecting the *C. elegans lmn-1* variant strains to a series of cellular and physiological assays, we could discern the degree of severity of different disease-causing human *LMNA* variants and assess the likelihood of VUS to contribute to disease. Our assays yielded a broad distribution of phenotypic severity within each *lmn-1* missense variant population, reflective of the wide range of symptom severity observed in human laminopathies. Previous *C. elegans* models show similar results, where animals expressing extra copies of GFP-tagged LMN-1 Y59C or LMN-1 L535P in an otherwise wild-type LMN-1 background have variously impaired movements [31,80]. Our data suggest that viability and motility are primary indicators of variant pathogenicity, followed by nuclear blebbing and nuclear migration defects. We established a scoring system which incorporates cellular and physiological defects in the *C. elegans* models to predict the potential pathogenicity of each variant (Table 1). Variant strains that were homozygous inviable scored two, and those with significantly reduced viability compared to wild type scored one. Variant strains where greater than 30% of the animals had swimming defects scored two, while strains in which less than 30% of animals exhibited swimming defects were scored zero. Finally, a score of one was assigned to strains with nuclear migration defects. By this metric (Table 1), we found that our

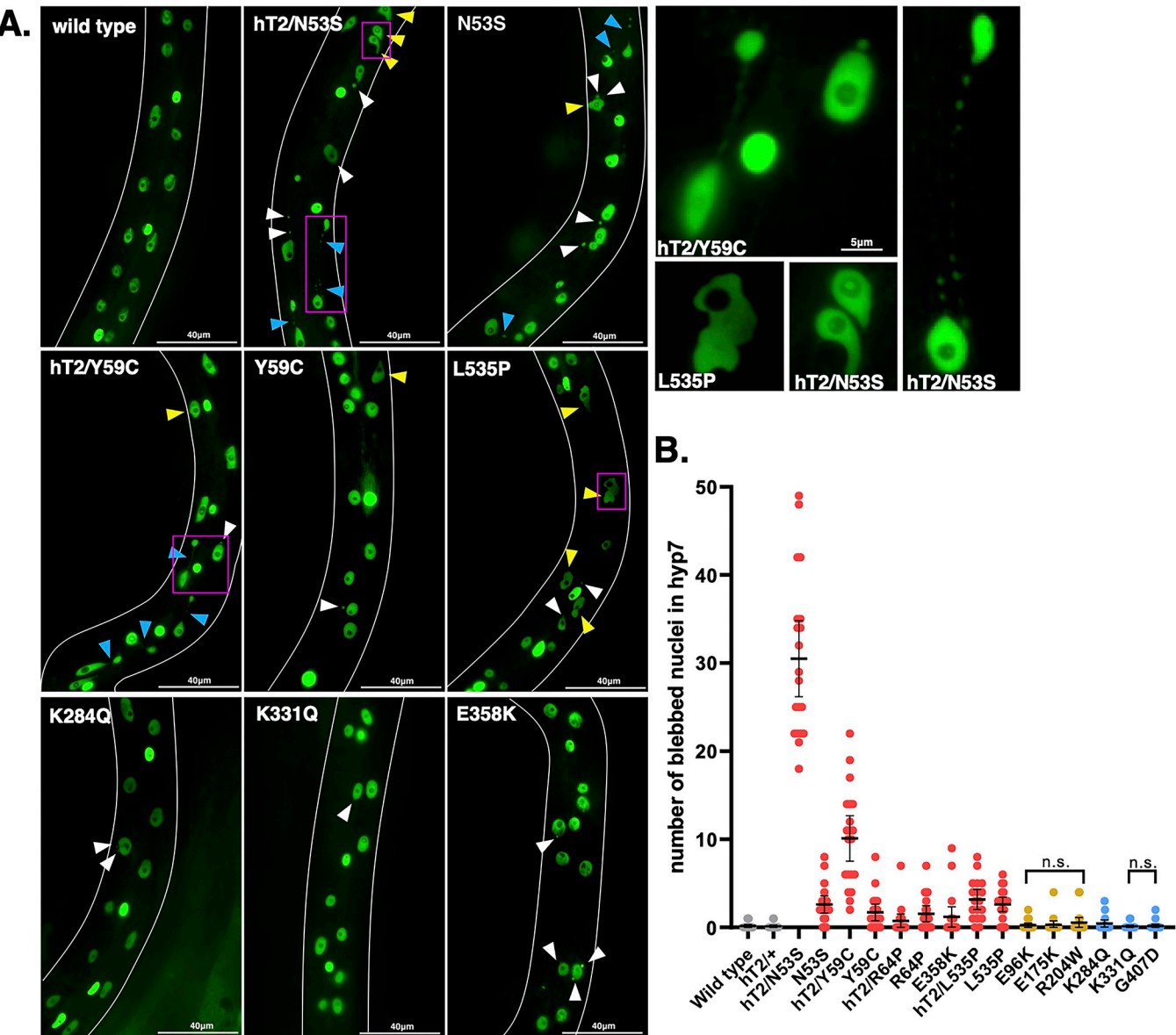

**Fig 6. Missense Mutations in *lmn-1* Often Result in Abnormal Hypodermal Nuclear Morphology and the Formation of Nuclear Blebs.** (A) Representative epifluorescence images of animals with the indicated genotypes expressing a SV40nls::GFP::LacZ soluble nucleoplasmic marker from UD398 in hyp7 cells. Nuclear blebs (white arrowheads), abnormally shaped nuclei (yellow arrowheads), and chromosome bridges (blue arrowheads) are indicated. Scale bar: 40μm. Magenta boxes designate nuclei featured in the insets on the right. Scale bar: 5μm. (B) Number of nuclear blebs on one lateral side of the hyp7 syncytia in a young adult. The genotypes are colored as in Fig 1. n = 20. Means and error bars of 95% CI are shown. To control for type 1 error, p values were adjusted using the Benjamini-Hochberg procedure with a false discovery rate of 5%. hT2 designates the presence of a balancer chromosome with a wild-type copy of LMN-1. All the genotypes are statistically significantly different from wild type, except the hT2/+ control and the five marked n.s. for not significant.

models reflect the distinction between severe *LMNA* mutations that affect both cardiac and skeletal muscle (score ≥ 2), and those that tend to contribute to cardiac defects alone (score <2). These scores correlate with the frequency that these variants are found in humans (Fig 1A). Childhood-onset *LMNA* variants are rare in comparison to adult-onset pathologies that are more likely to be passed through families. Finally, in agreement with previous predictions [68], our scoring system predicts that one *LMNA* VUS (p.R331Q) is pathogenic, while the others (p.K270Q, p.S407D, and p.G523R) are unlikely to contribute to disease.

**Table 1. Phenotypic Scoring of *C. elegans* Laminopathy Models.**

| Human *LMNA* | *C. elegans lmn-1* | Viability* | Swimming† | Nuclear migration§ | Score |
|---|---|---|---|---|---|
| N39S | N53S | 2 | 0 | 1 | 3 |
| Y45C | Y59C | 1 | 2 | 1 | 4 |
| R50P | R64P | 1 | 2 | 0 | 3 |
| E358K | E358K | 1 | 0 | 0 | 1 |
| L530P | L535P | 2 | 2 | 0 | 4 |
| E82K | E96K | 0 | 0 | 0 | 0 |
| E161K | E175K | 0 | 0 | 0 | 0 |
| R190W | R204W | 0 | 0 | 1 | 1 |
| K270Q | K284Q | 0 | 0 | 0 | 0 |
| R331Q | K331Q | 0 | 2 | 0 | 2 |
| S407D | G407D | 0 | 0 | 0 | 0 |
| G523R | G528R | 0 | 0 | N/A | 0 |

*Score 2 for homozygous inviable, 1 for decreased viability, and 0 for wild type

†Score 2 when >30% of animals show a severe defect, and 0 when <30% of animals have severe defects

§Score 1 for defective nuclear migration and 0 for wild type

Human striated muscle laminopathies are primarily dominant disorders [7]. We therefore included heterozygous animals in our analyses of severe *lmn-1* mutations that model both cardiac and skeletal defects (N53S, Y59C, R64P and L535P). These lines largely acted in a dominant fashion in our assays. For example, in our motility assay, Y59C, R64P, and N53S heterozygous and homozygous strains had similar average swimming rates. In contrast, N53S/+ and Y59C/+ animals had more severe nuclear envelope defects in comparison to their homozygous counterparts (Fig 6B). Higher numbers of nuclear blebs in heterozygous animals may result from a dominant-negative effect of wild-type LMN-1, as is hypothesized to occur in human *LMNA*-associated pathologies [16,81]. Similar nuclear morphology defects were observed in heterozygous and homozygous N53S, Y59C, and L535P *lmn-1* hypodermal nuclei, indicating a general loss of the nuclear structure integrity.

Nuclear instability can be a hallmark of striated muscle disease [82]. The presence of micronuclei, chromatin bridges, and nuclear blebs has been reported in mammalian tissue culture models of laminopathies [77,78]. We saw similar nuclear morphology defects, again suggesting *C. elegans* is a good model for studying laminopathies. The formation of nuclear blebs may originate from errors in chromosome segregation, as has been previously described in *lmn-1 (RNAi)* early embryos [22]. This idea is supported by decreased fertility in these *lmn-1* missense variant strains. Alternatively, human laminopathies that affect both skeletal and cardiac muscle often have dysmorphic nuclei, which are proposed to be the result of the extreme mechanical forces these nuclei are subjected to [83–85]. A third possibility is that nuclear deformation could result from altered nucleocyto-skeletal coupling in *lmn-1* missense variants, similar to what has been reported in other laminopathy models [31,86,87].

Lamins directly bind to LINC complexes through the nucleoplasmic domains of SUN proteins at the inner nuclear membrane [73,88]. LINC complex components have been implicated in DCM-CD and EDMD, but the mechanisms by which LINC interfaces with lamins and how defects in this interaction contribute to disease are poorly understood [15]. We found that cardiomyopathy-associated *lmn-1(R204W)* variant animals exhibited a strong nuclear migration defect but did not have defects in viability or motility. Furthermore, LINC complex components failed to localize to the nuclear envelope in *lmn-1(R204W)* animals. Thus, we have

isolated a separation-of-function allele that does not disrupt the global function of lamins. Instead, our data are consistent with a hypothesis that *lmn-1(R204W)* specifically disrupts the physical interaction between lamins and the LINC complex. This allele will be valuable for future experiments studying the role of LINC complexes in laminopathies and suggests that a potential mode of pathogenicity in R204W variant animals is through disruption of the interaction between SUN and LMN-1.

In summary, our data provides a blueprint for creating and evaluating *C. elegans* clinical avatars for laminopathy-associated missense variants of human *LMNA* and new models to understand the mechanisms of lamin function during normal development. We created molecular and physiological assays for identifying potentially pathogenic missense variants and used these indicators to develop a scoring index in our *C. elegans* models that can be used to predict the tissues affected and relative rate of disease progression in humans with orthologous variants. Furthermore, we identified a separation-of-function point mutation that is predicted to disrupt the interaction between lamins and LINC complexes. In the future, these new *C. elegans lmn-1* missense variant models could also be used to screen candidate drugs to treat striated muscle laminopathies.

## Materials and methods

### *C. elegans* genetics and gene editing

Strains were grown on nematode growth medium (NGM) plates spotted with OP50 *Escherichia coli* and maintained at room temperature (22–23˚C) [89]. Strains used in this study are listed in Table 2. Some strains were obtained from the Caenorhabditis Genome Center, which is funded by NIH Office of Research Infrastructure Programs (P40 OD010440).

*lmn-1* missense strains were generated using a *dpy-10* co-CRISPR strategy [90–93]. crRNAs and repair template sequences were designed using CRISPR optimal target finder (Gratz et al 2014 Genetics) and CRISPRscan (Moreno-Mateos et al 2015 Nature Methods) to minimize off-target effects and are in Table 2. Injection mixes were made and injected as described previously [94]. Briefly, for UD796, UD750 and UD788 missense strains, an injection mix containing 0.2µL *dpy-10* crRNA (0.6mM from Horizon discovery/Dharmacon), 0.5µL *lmn-1* crRNA (0.6mM), 2.47µL of universal tracrRNA (0.17mM), was combined and added to 7.68µL of purified Cas9 protein (0.04mM from UC Berkeley QB3). The assembly was completed with the addition of 0.28µL of the *dpy-10* single-stranded DNA oligonucleotide (ssODN) (500ng/µL) and 0.21µL of the *lmn-1* ssODN (500ng/µL) repair templates to form CRISPR-Cas9 complexes *in vitro*. For NMX419, NMX554, NMX563, NMX567, NMX589, NMX421, NMX557, and UD837 strains, injection mixes containing 0.8µL *dpy-10* crRNA (30pmol/µl from Synthego Corporation) and 1.65µL of each of two *lmn-1* crRNA per strain (30pmol/µl) was combined and added to 1µL of purified Cas9 protein (5µg/µl from PNA Bio, Inc). Following addition of 1µL of the *dpy-10* ssODN (500ng/µL) and 1µL of the *lmn-1* ssODN (500ng/µL) repair templates (Integrated DNA Technologies, Inc). Mixes were injected into the gonads of young adult hermaphrodites. The sequences of the crRNA and the ssODN repair templates are listed in Table 3. Animals were screened for successful edits using PCR primers ods2621 (5'-TGGCTCAACGCTTCTAGAAACTTC-3') and ods2672 (5'- CATGACAACCTACGCCA AGCAG-3') and all variants were validated by Sanger sequencing.

### Physiological and molecular assays

To score brood size, L4 animals were singled onto OP50 *E. coli* NGM plates and left to grow for 42 hours, during which they reach adulthood (18 hours) and lay their first batch of eggs (24 hours). Every 24 hours, worms were transferred to new plates, for a total of three days. The

**Table 2. Strains Used in this Study.**

| Strain | Genotype | Reference |
|---|---|---|
| N2 | wild type | [89] |
| UD796 | lmn-1(yc107[N53S]) I/hT2 [bli-4(e937) let(q782) qIs48[P<sub>myo-2</sub>::gfp; P<sub>pes-10</sub>::gfp; P<sub>ges-1</sub>::gfp]] (I; III) | This study |
| UD750 | lmn-1(yc96[Y59C]) I/hT2 (I;III) | This study |
| UD788 | lmn-1(yc105[R64P]) I/hT2 (I;III) | This study |
| NMX419 | lmn-1(tgx395[E96K]) I | This study |
| NMX554 | lmn-1(tgx526[E175K]) I | This study |
| NMX563 | lmn-1(tgx535[R204W]) I | This study |
| NMX567 | lmn-1(tgx539[K284Q]) I | This study |
| NMX589 | lmn-1(tgx561[K331Q]) I | This study |
| NMX421 | lmn-1(tgx397[E358K]) I | This study |
| NMX414 | lmn-1(tgx390[G407D]) I | This study |
| NMX557 | lmn-1(tgx529[G528R]) I | This study |
| UD837 | lmn-1(tgx550 [L535P]) I/hT2 (I;III) | This study |
| LW905 | lmn-1(tm1502) I/hT2 (I;III) | [71] |
| UD398 | him-8(e1489) IV; ycIs10[P<sub>col-10</sub>::nls::gfp::lacZ] V | [73] |
| UD899 | lmn-1(yc107[N53S]) I/ hT2 (I;III); him-8(e1489) IV; ycIs10V | This study |
| UD883 | lmn-1(yc96[Y59C]) I/hT2 (I;III); him-8(e1489) IV; ycIs10 V | This study |
| UD895 | lmn-1(yc105[R64P]) I/hT2 (I;III); him-8(e1489) IV; ycIs10 V | This study |
| UD915 | lmn-1(tgx395[E96K]) I; him-8(e1489) IV; ycIs10 V | This study |
| UD913 | lmn-1(tgx526[E175K]) I; him-8(e1489) IV; ycIs10 V | This study |
| UD908 | lmn-1(tgx535[R204W]) I; him-8(e1489) IV; ycIs10 V | This study |
| UD835 | lmn-1(tgx539[K284Q]) I; him-8(e1489) IV; ycIs10 V | This study |
| UD834 | lmn-1(tgx561[K331Q]) I; him-8(e1489) IV; ycIs10 V | This study |
| UD839 | lmn-1(tgx[E358K]) I; him-8(e1489) IV; ycIs10 V | This study |
| UD836 | lmn-1(tgx390[G407D]) I; him-8(e1489) IV; ycIs10 V | This study |
| UD857 | lmn-1(tgx550[L535P]) I/hT2 (I;III); him-8(e1489) IV; ycIs10 V | This study |
| BN147 | emr-1(gk119) I; bqSi142 [P<sub>emr-1</sub>::emr-1::mCherry] II | [98] |
| UD453 | unc-84(yc23[unc-84::GFP]) X | [99] |
| UD1008 | unc-84(yc23[unc-84::GFP]) X; emr-1(gk119)? I; bqSi142 [P<sub>emr-1</sub>::emr-1::mCherry] II | This study |
| UD1009 | unc-84(yc23[unc-84::GFP]) X; lmn-1(tgx535[R204W]), emr-1(gk119)? I; bqSi142 [P<sub>emr-1</sub>::emr-1::mCherry] II | This study |
| UD1023 | hT2 (I;III); him-8(e1489) IV; ycIs10V | This study |
| UD1024 | lmn-1(tm1502)/hT2 (I;III); him-8(e1489) IV; ycIs10V | This study |

total number of eggs and offspring on the previous plate was counted 24 hours after the animals were moved to new plates. The percent lethality of each strain was found by quantifying the number of unhatched eggs and dividing the sum by the brood size. The raw data are shown in S1 Data.

To assay swimming motility, 8–10 L4 animals were picked onto an unspotted NGM plate, which was then flooded with M9 buffer and placed on a light microscope equipped with a Samsung Galaxy A51 5G smartphone attached to the eyepiece. Videos were taken with the smartphone's camera at 30fps. Animals were filmed for thirty seconds. Videos were converted to AVI format using ffmpeg and subsequently processed for background subtraction and binarization in Fiji [95]. The number of body bends per second (BBPS), was quantified using the wrMTrck plugin [96]. The raw data are shown in S1 Data.

**Table 3. crRNA and Repair Templates Used in this Study.**

| New alleles | Strain | crRNA* | DNA repair template*†§ |
|---|---|---|---|
| lmn-1(yc107 [N53S]) I/ hT2 [bli-4 (e937) let (q782) qIs48 [P_myo-2::gfp; P_pes-10::gfp; P_ges-1::gfp]] (I;III) | N2 | AAAACUCACGUCGAUGUAAG | GGCTCAACGCTTCTAGAAACTTCACGTCTCAAGAGAAGAATCATTTGACGTCACTCAgCAGTCGTCGTCTTGCtACTTACATCGACGTGAGTTTAATTTGAAAGTATTCATATTTGA |
| lmn-1 yc96 [Y59C] I/ hT2 (I;III) | N2 | AAAACUCACGUCGAUGUAAG | AAGTGTACTTTCAAATATGAATACTTTCCAAATTAAAACTCACGTCGATGcAcGTaGCAAGACGACTGTTGAGTGAGTCAAATGATCTTTCTCTTGAAGACGTGA |
| lmn-1(yc105 [R64P]) I/ hT2 (I;III) | N2 | GCAAGAGAACAACAGACUCC | TTTTAATTTGGAAAGTATTCATATTTGAAAGTACACTTTTCAGAAAGTTCcTCAAcTGGAGCAAGAGAACAACAGACTCCAaGTTCAAATTCGCGACATCGAAGTTGTTGAAAAGAAAGAAGGAGTCAAACTT |
| lmn-1 (tgx395 [E96K]) I | N2 | CGAUGUCGCGAAUUUGAACC / CGAUCGCTTCGAGGCGGAAA | CGTCAATTGGAGCAAGAGAACAACAGACTCCAGGTcCAgATcCGttGAtATtGAgGTcGTcGAgAAaAAgGAaAAaTCtAAtTTagCCtGAcCGtTTCaAGGCtGAAAAGGTACACTTTGTTTATATTCTGATCGCCAAA |
| lmn-1 (tgx526 [E175K]) I | N2 | GGCACGCAACGAUAAAUTGG / AGCCGCCAACAATAAAATCA | AAAACAGAGAGACGTTGCAGGCACGCAACGATAAATTagTTtGTtaAGAAcGAcACTtAAgAAGCAaAACATtACcCTTCGcGAtACCGTcGAgGAtGAtcAgCCtTCtCGcACcGCtGCtAAcAAcAAgAATcAAGGCTCTGGAAGAAGATCTCGCTTTGCTCTT |
| lmn-1 (tgx535 [R204W]) I | N2 | ACTCTTCGTGACACCGTAGA / TTTGCTTCTCAACAGCACAA | CTCAAAAAGCAGAACACTCACTCTTCGTGACACCGTtGAgGGACTtAAaAAgGCtGTTGAgGAcGAgACTCTCCTCTCGGACAGCTGCCAAtAAcAAgAATtAAGGCCtTtGAgGAgGAcCTtGCcTTTGCcCTccAgCAaCACCAAGGGAGAACTTGAGAGAAGTTCGTCACAAGAGAC |
| lmn-1 (tgx539 [K284Q]) I | N2 | AGCATTGAGCTTGTTTTTGT / GACTTGGAGACATCAAGCAG | GCATCAAAACAAAACAGCTTTCGAAGATGCCTACAgAgACcAGCTtAAcGCcGCcGCCGcCGTGAgCGtCAgGAaGAaGGCGGTCGTCCtCAGGACGCcCATtCAcCTcaGAGCCaGaGTTcaGAGACCTcGAaAccTCcAGcAGTGGAAAATGCTTCGCTCATCGAACGTCTTCGTT |
| lmn-1 (tgx561 [K331Q] I | N2 | GACTTGGAGACATCAAGCAG / CCAAGAGAAGCTCGACGACA | CTTCGTGCCCGTGTTCGTGACTTGGAGACATCAAGcAGcGGAAAcGCcCTcCTtATTcGAyaGaCTTaGaTCtGAacTTgAtACcCTccAacGtTCTtTCCAyGAaAAGCTtGAtGACAAGGATGCTCGAGTTGCTGAACTTAATCAAGAG |
| lmn-1 (tgx397 [E358K] I | N2 | CCAAGAGAAGCTCGACGACA / TCTTGAGTTCGGCGTCCAAT | CACTCTGAAGAGATCGGTTCCAAGAGAGAGCTCGACGATAAAGAGGCcCGcATcGCcGAgtTgAAcCAgGAaATtGAaCGaATGAtGtlcCaAGTTtCAtcGAcTTgtTgGAcGTcAAgATtCAgTTGGACGCGCGAACTCAAGACACTACCAAGCTCTCCTTG |
| lmn-1 (tgx390 [G407D]) I | N2 | GCGTCTCAATCTTACTCAGG / GGAGGAGCCAAGCGCGTCAGCG | CCTTGAGGGTGAGGAGGAGCCGTCTCAATCTTACTCAyAAGAGGCaACCgCAgAaAtACcTCgGTcCAcCATcGTtTCcTTcTTCgTCCGatGGcGGcGcctcCGCcCAGCGCGGAGTGAAGCGTCGTCGGTTGTCGATGTAA |
| lmn-1 (tgx529 [G528R]) I | N2 | AGCATCCGAGACCAAACGG / GCTCGTCTTGAGGATAGTGA | TCGTATGAAGCTCGTCCACATGCTAGCGCCACCGTCtGGTCcGCtGAcGCcGGaGCcGagGGTtTAcGTcATcGAaAAgCchgCAaTGGCCtATccGtGAcAAtCCATCtGCcCGcCtCGGAGAcAGTGAGGAGACACTGTTCTTCTATCACCGTTGAAT |
| lmn-1 (tgx550 [L535P]) I/ hT2 (I;III) | N2 | CTATCCTCAAGACGAGCTGA / GCTCGTCTTGAGGATAGTGA | ACAATTTTCAGTGGCCAATTGAGATAACCCATCAGCccCGcCcaGAGGActcTGAAGGAGACACTGTTTCTCTATCACCGTTGAAT |

---

* All nucleotide sequences are displayed in 5' to 3' orientation.

† Lowercase nucleotides differ from the genomic sequence and include the missense, PAM site, and synonymous screening mutations.

§ Underlined sequences indicate the missense mutation.

Nuclear migrations of embryonic hyp7 precursors were scored in larval animals by counting the number of nuclei abnormally localized in the dorsal cord as described previously [74]. To assay nuclear morphology, L4 animals were picked onto plates and allowed to grow for 20 hours. Young adults were then mounted onto 2% agarose pads in ~5µL of 1mM tetramisole in M9 buffer. The number of nuclear blebs were counted, including those associated with chromosome bridges. Only one lateral side of each animal was scored. Nuclei were visualized using a wide-field epifluorescent Leica DM6000 microscope with a 63 × Plan Apo 1.40 NA objective, a Leica DC350 FX camera, and Leica LAS AF software. The raw data are shown in S1 Data.

### *lmn-1(R204W)* immunofluorescence and imaging

Comma-stage wild-type and *lmn-1(R204W)* embryos were fixed and stained with monoclonal antibody 1209D7 against UNC-83c as previously described [75]. The anti-LMN-1 polyclonal guinea pig antibody was used at a dilution of 1:1000 (gift of Jun Kelly Liu, Cornell University) [97]. Fluorescence intensity of dorsal hypodermal nuclei was calculated using the following equation: Corrected total nuclear fluorescence = Integrated density–(Area of each nucleus x Mean fluorescence of the background). The integrated density is the product of the selected area and the average gray value within the selection. Values were found using ImageJ. Confocal images were taken on a Zeiss LSM 980 with Airyscan using a 63x Plan Apo 1.4 NA objective and the Zeiss Zen Blue software made available through the MCB light imaging microscopy core and through NIH grant S10OD026702. The raw data are shown in S1 Data.

### Statistics

Scatter plots show the mean and 95% confidence intervals (CI) as error bars. Swimming phenotypes were analyzed using a chi-squared test comparing the number of wild type animals that fell below a threshold of 1.1 BBPS, which corresponds to the corroborated mean swimming rate of *lmn-1(Y59C)* worms [32,72] to the missense mutant populations. Percentages and their corresponding p values are shown in the bar graph. Unpaired student's t tests were used for nuclear migration and were corrected with Tukey. Nuclear morphology was evaluated using the Benjamini-Hochberg adjusted p value and setting a false discovery rate of 0.05. Graphs were generated with Prism 9 software.

### Supporting information

**S1 Data. Data from phenotypic assays.** The raw data are shown for each of the quantitative assays. Each column is labeled by the *C. elegans* LMN-1 mutant assayed. The data are presented in sections. Data for Fig 2A show the brood size per animal. Data for Fig 2B are the percentage of eggs laid that hatched per mother. Fig 3B and 3C data are the BBPS in individual animals over a 30 second video. Data for Fig 4C are the number of hyp7 nuclei abnormally found in the ventral cord of L1-L2 animals, a proxy for failed nuclear migrations. Fig 5B and 5C show the intensity of anti-UNC-83 or anti-LMN-1 immunostaining, respectively. Values are in arbitrary units and each number represents a single hyp7 nucleus from a comma stage embryo. Data for Fig 6B are the numbers of hypodermal nuclei with blebs per side of an adult animal. See the individual figure legends for more details.
(XLSX)

**S1 File. Wild-type Animals Exhibit Robust Swimming Behavior.** An example video of L4 stage *C. elegans* swimming in buffer for 30 seconds. The number of body bends is shown in gray text next to each animal and was generated by the Fiji wrMTrck plugin.
(MP4)

**S2 File. Homozygous *lmn-1(Y59C)* Animals have Impaired Motility.** A representative video of homozygous *lmn-1(Y59C)* L4 stage animals thrashing in buffer for 30 seconds. The number of body bends is shown in gray text next to each animal and was generated by the Fiji wrMTrck plugin.
(MP4)

**S3 File. Homozygous *lmn-1(R64P)* Animals Demonstrate Swimming Behavior Ranging from Normal to No Motility.** An example video of homozygous *lmn-1(R64P)* L4 stage animals swimming in buffer for 30 seconds. The number of body bends is shown in gray text next to each animal and was generated by the Fiji wrMTrck plugin.
(MP4)

**S4 File. A VUS, *lmn-1(K331Q)*, Significantly Reduces Swimming Motility in *C. elegans*.** A representative video of *lmn-1(K331Q)* L4 animals thrashing in buffer for 30 seconds. The number of body bends is shown in gray text next to each animal and was generated by the Fiji wrMTrck plugin.
(MP4)

**S5 File. Homozygous *lmn-1(L535P)* animals have the most severe swimming defect.** An example video of homozygous *lmn-1(L535P)* animals swimming in buffer for 30 seconds. The number of body bends is shown in gray text next to each animal and was generated by the Fiji wrMTrck plugin.
(MP4)

## Acknowledgments

We thank members of the Starr/Luxton lab for helpful discussions, editing of the paper, and technical help, especially Jamie Ho and Daniel Elnatan for help with imaging. We thank Thomas Wilkop at the MCB Light Microscopy Imaging Facility, which is a UC Davis Campus Core Research Facility, for microscopy assistance.

## Author Contributions

**Conceptualization:** Ellen F. Gregory, Gisèle Bonne, G. W. Gant Luxton, Christopher Hopkins, Daniel A. Starr.

**Data curation:** Ellen F. Gregory, Trisha Brock, Gisèle Bonne, Christopher Hopkins.

**Formal analysis:** Ellen F. Gregory, Trisha Brock.

**Funding acquisition:** Gisèle Bonne, G. W. Gant Luxton, Christopher Hopkins, Daniel A. Starr.

**Investigation:** Ellen F. Gregory, Shilpi Kalra, Trisha Brock.

**Methodology:** Ellen F. Gregory, Shilpi Kalra, Trisha Brock.

**Project administration:** Ellen F. Gregory, Christopher Hopkins, Daniel A. Starr.

**Resources:** Ellen F. Gregory, Shilpi Kalra, Trisha Brock, Gisèle Bonne.

**Supervision:** G. W. Gant Luxton, Christopher Hopkins, Daniel A. Starr.

**Validation:** Ellen F. Gregory.

**Visualization:** Ellen F. Gregory.

**Writing – original draft:** Ellen F. Gregory, Christopher Hopkins, Daniel A. Starr.

**Writing – review & editing:** Ellen F. Gregory, Shilpi Kalra, Trisha Brock, Gisèle Bonne, G. W. Gant Luxton, Christopher Hopkins, Daniel A. Starr.

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
