## [Decision Letter · Decision Letter 0]

16 May 2023

Dear Dan,

Thank you very much for submitting your Research Article entitled 'Caenorhabditis elegans models for striated muscle disorders caused by missense variants of human LMNA' to PLOS Genetics.

The manuscript was fully evaluated at the editorial level and by two independent peer reviewers. The reviewers appreciated the attention to an important topic but identified some concerns that we ask you address in a revised manuscript.

As you will see, the reviewers appreciated the importance of the topic and found your study to be well designed, technically sound and of interest to the field. They did, however, raise several concerns that should be addressed prior to further consideration, and I also added a couple of suggestions/questions. In particular, please note the following:

1. The reviewers appreciated the use of CRISPR/Cas to generate mutant alleles expressed at physiological levels. However, given the possibility of off-target mutations, Reviewer #2 wondered whether and how many times these strains were out-crossed. Alternatively, if you used several independently isolated mutant strains and they all had the same phenotype, that would also address this concern.

2. Reviewer #2 noted that the wild type controls used for comparison with the homozygous mutants were themselves not derived from balanced strains, and reviewer #1 also wondered about a possible contribution form the balancer. In this regard, future readers of this study will likely include individuals who are not familiar with the concept of balanced strains. It might be helpful to explain how this works, and specifically the effect of maternally contributed wild type protein on to the ability of the mutant strains to reach adulthood.

3. Both reviewers noted that the study does not adequately address the issue of micronuclei, and Reviewer #1 also wondered whether the nucleoplasmic marker you used is sufficient for detecting micronuclei, blebbing and chromosome bridges.  

4. As *C. elegans* has only a single lamin gene, a more detailed discussion on its similarities to Lamin A/C vs. Lamin B is warranted, as noted by Reviewer #1. This is especially relevant because you are modeling Lamin A/C mutations in LMN-1, which has features that resemble Lamin B.

5. In Figure 5, it is not clear why only the one mutant was analyzed, when others had a nuclear migration defect? Specifically, you attribute the migration phenotype to a possible defect in LMN-1::LINC interaction, but without checking several lmn-1 mutants that do or do not display a migration defect this correlation is premature.

6. The comments regarding statistical analyses, as noted by Reviewer #1, should be addressed. It might help is you explained why different tests were applies in the Figures with multiple comparisons to the wild type sample.

7. A few statements should either be elaborated on or toned down, as suggested by both reviewers.

8. Why does Table 1 not include the micronuclei phenotype? That would “elevate the status” K284Q allele.

1) Provide a point-by-point list of your responses to the comments listed above, the additional review comments, and a description of the changes you have made in the manuscript.

We hope to receive your revised manuscript within the next 30 days, and we anticipate being able to make a decision without another round of reviews. If you anticipate any delay in its return, we would ask you to let us know the expected resubmission date by email to plosgenetics@plos.org.

Yours sincerely,

Orna Cohen-Fix

Guest Editor

PLOS Genetics

Gregory P. Copenhaver

Editor-in-Chief

PLOS Genetics

Reviewer's Responses to Questions

**Comments to the Authors:**

Reviewer #1: The manuscript by Gregory and colleagues describes the phenotypes of a series of C. elegans mutants designed to mimic missense mutations in the human LMNA gene. LMNA encodes lamin A and lamin C proteins that are components of the nuclear lamina in all differentiated cell types. A remarkably high number of LMNA mutations are causatively linked to a diverse set on human diseases known as laminopathies. Despite intensive research efforts, it is largely unknown how the individual changes in lamin A (and lamin C) lead to clinical manifestations in specific tissues. In this manuscript, the authors focused on 5 autosomal dominant mutations that cause both cardiac and skeletal muscle defects, 3 mutations linked to cardiac muscle defects only and 4 mutations that the authors categorized as variants of unknown clinical significance (VUS). All 12 mutations are missense mutations that affect a single amino acid residue conserved between the human lamin A and C. elegans LMN-1 proteins: CRISPR/Cas9 genome engineering was used to introduce the relevant mutations into the endogenous lmn-1 gene.

The 5 mutations in the first category caused a severe reduction in viability and fertility, whereas the mutations in the two other categories had no or little effect on these parameters. To evaluate muscle activity, worms were placed in liquid medium and body bends per second were measured. This revealed reduced activity for 3 of the mutants in the first category and for 1 of the VUS. Nuclear migration defects were observed for 3 mutants; delocalization of LINC (Linker of the Nucleoskeleton and Cytoskeleton) complex members was evaluated and observed for 1 of these mutants. Finally, nuclear morphology was found to be abnormal in several mutants. Combining the data on viability, swimming and nuclear migration a score was assigned to each mutant. Although the mutants show variability within each parameter, the combined score separated the 5 severe mutations (causing cardiac and skeletal muscle defects in humans) from the remaining mutations.

The manuscript is technically very sound, well written and will be of interest to many colleagues studying nuclear organization, laminopathies and/or muscle function. The modification of the endogenous lmn-1 represents an important improvement compared to most previous studies. However, I suggest the authors to consider the following points prior to publication:

Some claims seem unjustified. For instance, “we uncovered molecular mechanisms for how lamins interact with other nuclear envelope proteins to carry out their cellular functions” and “The modeled missense mutations also uncovered new mechanistic insights into the normal roles of lamin in development.” It is not clear to me which are the novel molecular mechanisms.

The manuscript should describe the differences between A and B type lamins. Since the first publication describing its role in C. elegans, LMN-1 has been termed a B-type lamin. According to Uniprot, the identity between LMN-1 (Q21443) and LMNB1 (P20700) is 31.3% whereas the identity between LMN-1 (Q21443) and LMNA (P02545) is 29.4%. Like B type lamins in humans (and other vertebrates), LMN-1 is presumably permanently farnesylated, whereas lamin A proteins is posttranslationally modified to remove the farnesylated C terminus. This does not necessarily influence the conclusions of the manuscript but the authors should explain to the readers that the expression and processing of lamins are more complex in humans as compared to C. elegans.

The information in lines 101-106 should be updated to mention that Penfield and colleagues expressed and evaluated an un-tagged, single-copy LMN-1(N209K) mutant in a lmn-1(0) background (doi: 10.1091/mbc.E17-06-0374).

One on the main arguments by the authors is that C. elegans is an attractive model to evaluate VUS. Of the 4 putative VUS tested in the manuscript, R331Q showed a swimming defect. However, based on references 60 and 66, is R331Q indeed a VUS? Ref 66 concluded that “Clinical, morphological, functional, haplotype, and segregation data all indicate that LMNA p.(Arg331Gln) is a pathogenic founder mutation” and in the ClinVar databased R331Q is listed as “ Pathogenic(8); Likely pathogenic(1); Uncertain significance(1)”. I suggest the authors to consider is R331Q classifies as a VUS. I think the argument that C. elegans is an attractive model to evaluate VUS compared to for instance vertebrate models in terms of cost, speed, bioethics, etc. is still equally valid.

The p values in Figure 2A-B should be corrected for multiple comparisons as in Fig3B. Similarly, which test was used in Figure 4C? Was correction for multiple comparisons applied?

The heading in line 192 should be corrected: only 4 of 12 had impaired swimming behaviour.

In several occasions the genotype-phenotype correlation is not straight forward. For instance, Y59C and L535P correspond to dominant mutations in humans. In the swimming assay, heterozygous and homozygous Y59C mutants behave similar whereas homozygous L535P mutants are more affected than heterozygous L535P mutants. In contrast, in terms of nuclear morphology, the Y59C mutation affects heterozygous animals more than homozygous individuals whereas heterozygous and homozygous L535P mutants behave similar. The behaviour of Y59C in latter assay is discussed superficially in lines 348-351. I think the authors should discuss this in greater detail. Could the balancer chromosome affect the nuclear morphology assay?

Also regarding the evaluation of nuclear morphology: The reporter should be described in more detail. Which NLS was used, how many copies, etc? Without a nuclear envelope marker, can the authors confidently distinguish between micronuclei (i.e. separated from main nucleus) or nuclear blebs when the GFP signal is close to the main nucleus? The potential presence of chromatin bridges without using a chromatin marker is not convincing.

The observations reported in Figure 5 are interesting, but why was only a single mutant analysed and not all three mutants with nuclear migration defects? By IF, there is less UNC-83 in the nuclear envelope of cells of R204W embryos whereas UNC-84::GFP seems completely absent in the hypodermis of R204W adults. Is the difference in behaviour of UNC-83 vs UNC-84 due to the method (IF vs live), the stage (embryos vs adult) or because they are different proteins? Finally, there seems to be less LMN-1(R204W) at the nuclear envelope. This would be relevant to document by quantification.

The legend to Figure 6B mentions p values but these are not represented in the Figure.

Reviewer #2: “Caenorhabditis elegans models for striated muscle disorders caused by missense variants of human LMNA” by Ellen F. Gregory et al.

In their manuscript, Gregory et al successfully employ CRISPR/Cas9 genome editing techniques to produce a variety of nematode lines, each containing different human LMNA mutations. These mutations span a wide range, from those known to cause muscle disorders to those whose clinical implications remain unclarified. Through the use of a comprehensive selection of established testing methods, applied at both the cellular and organismal levels, the authors convincingly demonstrate that these mutations give rise to diverse phenotypes. Importantly, these phenotypic expressions generally align with the known severity of the corresponding human diseases. Remarkably, certain mutations previously categorized as clinically ambiguous also yield significant phenotypes. This study thus provides a straightforward approach to potentially ascertain the clinical impact of newly discovered variants. Given its implications, it is my belief that this work is well-suited for publication in PLoS Genetics.

Major comments:

The major area of concern in this study relates to the controls used to account for genetic background. It remains unclear whether the animals that were engineered underwent any out-crossing to exclude the potential influence of nonspecific editing. Additionally, in all conducted tests, balanced mutants were compared directly with wild-type animals. However, these balanced mutants not only carry a single copy of the mutation but also contain the balancer construct. The potential impact of this balancer on the results should be addressed and verified. Without these clarifications, there might be an underlying risk of attributing observed phenotypes to the specific mutations, while they might be influenced by other genetic factors.

Minor comments:

1. While the schematic model in Figure 1C provides valuable insights, the manuscript would be significantly enhanced by the inclusion of a more detailed structural model that maps these mutations to LMN-1. Tools such as AlphaFold, among others, can be utilized to generate these intricate and realistic structural models with relative ease.

2. Previous studies have reported the occurrence of micronuclei in cell lines expressing specific LMNA mutations. Consequently, it is crucial for the manuscript's discussion section to address this observation. It should also evaluate and elucidate on the presence or absence of evidence pertaining to micronuclei formation in the chosen mutations for this study. Furthermore, an analysis highlighting the pros and cons of using nematodes over cell lines should be provided.

3. The scale bar at 3A appears to be wrong, as it suggests animals are <0.3mm long.

4. Figure 3C needs error/confidence bars

5. In the water motility assay, Y59C, R64P and N53S had similar averages in het and homozygous state. This should be explicitly mentioned and discussed, similarly to the way it was discussed for the micronuclei.

6. The statement, "However, animals that survive due to a presumably low level of RNAi," makes an assertion without robust mechanistic evidence to support it. A more measured speculation could be to suggest that survival is potentially due to increased residual protein levels.

7. How were the strains validated? If by Sanger sequencing, it should be mentioned in the methods section.

**Have all data underlying the figures and results presented in the manuscript been provided?**

Reviewer #1: Yes

Reviewer #2: Yes

PLOS authors have the option to publish the peer review history of their article (what does this mean?). If published, this will include your full peer review and any attached files.

Reviewer #1: No

Reviewer #2: **Yes: **Daniel Bar

---

## [Editor Report · Decision Letter 1]

1 Aug 2023

Dear Dr Starr,

After reading the revised manuscript, we are delighted to inform you that your manuscript, "Caenorhabditis elegans models for striated muscle disorders caused by missense variants of human LMNA" has been editorially accepted for publication in PLOS Genetics. Not only are your findings of interest to a broad audience interested in disease modeling and nuclear structure/function, but your approach, and in particular the considerations that went into the variants of unknown significance will be useful for others who wish to model disease mutations in model organisms. Congratulations!

A few very minor things:

1. In the editorial copy of the pdf, the text in Figure 1 was very pixilated. It could be a pdf conversion thing that is specific to the editorial process.

2. in lines 117, 119, 139 and 146 (and I may have missed a few), I think that it should be LMN-1 rather than lmn-1 because the amino acid substitutions refer to the protein, not the gene

3. There is an extra "as" in line 144.

Yours sincerely,

Orna Cohen-Fix

Guest Editor

PLOS Genetics

Gregory P. Copenhaver

Editor-in-Chief

PLOS Genetics

**Data Deposition**

http://datadryad.org/submit?journalID=pgenetics&manu=PGENETICS-D-23-00419R1

**Press Queries**

---

## [Editor Report · Acceptance letter]

18 Aug 2023

PGENETICS-D-23-00419R1 

* Caenorhabditis elegans * models for striated muscle disorders caused by missense variants of human * LMNA *

Dear Dr Starr, 

We are pleased to inform you that your manuscript entitled "* Caenorhabditis elegans * models for striated muscle disorders caused by missense variants of human * LMNA *" has been formally accepted for publication in PLOS Genetics! Your manuscript is now with our production department and you will be notified of the publication date in due course.

With kind regards,

Livia Kovacs

PLOS Genetics

On behalf of:
